# The impact of uncertainty in black carbon's refractive index on simulated optical depth and radiative forcing

Ruth A.R. Digby[1,2], Knut von Salzen[1,2], Adam H. Monahan[2], Nathan P. Gillett[1,2], and Jiangnan Li[1]

[1]Canadian Centre for Climate Modelling and Analysis, Environment and Climate Change Canada
[2]School of Earth and Ocean Sciences, University of Victoria, Victoria, British Columbia, Canada

**Correspondence:** Ruth A.R. Digby (ruth.digby@ec.gc.ca)

**Abstract.**

The radiative forcing of black carbon (BC) is subject to many complex, interconnected sources of uncertainty. Here we isolate the role of the refractive index, which determines the extent to which BC absorbs and scatters radiation. We compare four refractive index schemes: three that are commonly used in Earth system models, and a fourth more recent estimate with higher absorption. With other parameterizations held constant, changing BC's spectrally-varying refractive index from the least- to most-absorbing estimate commonly used in Earth system models ($m_{550nm} = 1.75 - 0.44i$ to $m_{550nm} = 1.95 - 0.79i$) increases simulated absorbing aerosol optical depth (AAOD) by 42% and the effective radiative forcing from BC-radiation interactions (BC ERFari) by 47%. The more recent estimate, $m_{532nm} = 1.48 - 0.84i$, increases AAOD and BC ERFari by 59% and 100% respectively relative to the low-absorption case. The AAOD increases are comparable to those from recent updates to aerosol emission inventories, and in BC source regions, up to two thirds as large as the difference in AAOD retrieved from MISR and POLDER-GRASP satellites. The BC ERFari increases are comparable to previous assessments of overall uncertainties in BC ERFari, even though this source of uncertainty is typically overlooked. Although model sensitivity to the choice of BC refractive index is known to be modulated by other parameterization choices, our results highlight the importance of considering refractive index diversity in model intercomparison projects.

## 1 Introduction

Black carbon (BC), formed as a result of incomplete combustion, is the most strongly warming of the aerosols (Szopa et al., 2021), with particularly important impacts in the Arctic (AMAP, 2021; Sand et al., 2016; von Salzen et al., 2022). Although its effective radiative forcing (ERF; Boucher et al., 2013) is understood to be positive, the magnitude of this forcing remains uncertain (AMAP, 2021; Bond et al., 2013; Szopa et al., 2021; Thornhill et al., 2021).

This uncertainty stems from many sources, including uncertainty in the optical properties of freshly emitted BC (Bond and Bergstrom, 2006; Bond et al., 2013; Liu et al., 2020) and the changes in these properties as BC ages (Hu et al., 2023; Peng et al., 2016; Taylor et al., 2020); in the total burden and vertical distribution of BC (Gliß et al., 2021; Samset et al., 2013; Sand et al., 2021), which are themselves complicated by uncertainties in BC emissions (Bond et al., 2013; Hoesly et al., 2018;

Wang et al., 2021) and lifetime (Gliß et al., 2021; Hodnebrog et al., 2014; Lund et al., 2018); and uncertainty in the details of
BC-cloud interactions (Koch and Del Genio, 2010; Szopa et al., 2021; Zanatta et al., 2023).

Constraining the climatic effect of BC is complicated by the fact that models and observations frequently disagree, but it is not always clear which – if either – is correct (Samset et al., 2018b). For example, models tend to simulate lower absorption aerosol optical depth (AAOD) than is measured by AERONET stations (Bond et al., 2013; Samset et al., 2018b; Sand et al., 2020). This can be interpreted to mean that models underestimate AAOD, perhaps because of too-low emissions or too-
vigorous removal (Bond et al., 2013) or because the simulated aerosols are insufficiently absorbing (Bond and Bergstrom, 2006; Liu et al., 2020). Alternatively it could indicate that AERONET overestimates AAOD, as suggested by Andrews et al. (2017) based on comparisons between AERONET and in-situ aircraft measurements. It is also possible that both models and observations are reasonably accurate, and that the apparent discrepancy comes from comparing point-source measurements of a very spatially heterogeneous quantity against model grid cell averaged fields (Schutgens et al., 2021; Wang et al., 2016, 2018).
In this work we investigate the sensitivity of BC AAOD and radiative forcing to one key source of uncertainty: the complex refractive index, which determines the degree to which an aerosol absorbs and scatters radiation. Lab-based estimates of the BC refractive index (BCRI) show remarkable diversity (Bond and Bergstrom, 2006; Liu et al., 2020), and this diversity is reflected in the range of values commonly used in Earth system models (Table 2). Recent studies (Sand et al., 2021; Gliß et al., 2021) suggest that uncertainty in the choice of BCRI may, along with uncertainties in the treatment of mixing and ageing, contribute
to the diversity in BC absorption simulated by Earth system models; however, no study has to date isolated the impact of BCRI on key BC fields in a single model with other aerosol treatments held constant. Here we select four BCRI schemes, three of which are commonly used in the climate modeling community and one more recent lab-based estimate which serves as an upper bound on the likely absorption of atmospheric BC, and use them to run otherwise-identical ensembles of simulations in the atmospheric model CanAM5.1-PAM. We contextualize the resulting spread in climate-relevant quantities including AAOD
and the effective radiative forcing from aerosol-radiation interactions (ERFari) by comparing with a number of other known uncertainties. In a companion analysis, Li et al. (2024) use an offline radiative transfer model to examine the sensitivity of wavelength- and mixing-state-dependent optical properties and radiative effects of BC to the choice of BCRI. Taken together, these works illustrate the influence of uncertainty in the BCRI on BC's simulated climate impacts.

## 2   The Refractive Index of Black Carbon

The refractive index $m = n - ik$ is a complex, wavelength-dependent parameter that determines the extent to which an aerosol absorbs or scatters radiation. It cannot be measured directly but is inferred by fitting laboratory measurements to an assumed optical model which describes the optical properties of an aerosol as a function of refractive index. For BC, these may be measurements of the scattering and absorption of light by flame-generated BC particles or of the reflectance at different angles by a compressed BC sample (Bond and Bergstrom, 2006). The measurements are then inverted to yield the full spectrally-varying
refractive index, for example using the Kramers-Kronig relations (Chang and Charalampopoulos, 1990) or the Drude-Lorentz dispersion relation (Dalzell and Sarofim, 1969; Lee and Tien, 1981). The former method is exact, but requires measurements

over a greater range of wavelengths; the latter requires fewer measurements but yields poor results at visible wavelengths (Chang and Charalampopoulos, 1990; Bond and Bergstrom, 2006; Menna and D'Alessio, 1982).

The choice of optical model can introduce substantial uncertainty into the derived refractive index. Historically, many estimates of the refractive index of black carbon have used Mie theory (Mie, 1908) which provides an exact solution to Maxwell's equations for scattering from spherical particles. However, freshly emitted BC particles are not spherical, but instead consist of fractal-like aggregates of individual monomers. Assuming Mie theory can result in substantial underprediction of these particles' absorption and scattering; furthermore, the inferred BCRI describes a combination of pure BC and the air contained within the aggregates' voids, whereas the objective is to measure the refractive index of pure BC (Bond and Bergstrom, 2006). Improving on Mie theory, a number of optical models for aggregate particles exist. Rayleigh-Debye-Gans (RDG) theory remains the most frequently used due to its simplicity (Liu et al., 2020), but it provides an approximate solution only and does not account for multiple scattering between the monomers, which may lead to underestimation of the mass absorption cross section (Mackowski, 1995; Bond and Bergstrom, 2006). Numerically exact solutions for aggregate particles include the multisphere T-matrix (MSTM; Mackowski and Mishchenko, 1996) and generalized multi-particle Mie (GMM; Xu, 1995) methods for aggregates composed of non-overlapping spheres, or the discrete dipole approximation (DDA; Purcell and Pennypacker, 1973; Yurkin and Hoekstra, 2007) for more complex morphologies. For more on these methods, the interested reader is referred to Kahnert and Kanngießer (2020). In practice, however, the RDG approximation is often sufficient for BC since its scattering is so low and other uncertainties are so high (Kahnert and Kanngießer, 2020; Mackowski, 1995). The four BCRI schemes assessed in this work were derived using Mie theory or RDG-type optical models.

For this study, we select four representative BCRI schemes from the literature: three that span the range of BCRI commonly used in climate models, and one more recent laboratory estimate that serves as an upper bound on the likely absorption of atmospheric BC. We use the term *scheme* both to emphasize the fact that the BCRI is not a single value but rather varies with wavelength, and also because we co-vary the density of BC with its refractive index. This decision is made for two reasons. The first is physical consistency: in order to derive the BCRI from optical measurements, one must assume a value for the density, and so the BCRI scheme is conditional on that chosen value. The second reason is modeling convention: although modeling centres may tune the density and refractive index independently it is generally true that models use either a low-absorption BCRI and high density, or vice versa, in order to obtain reasonable estimates of BC absorption. For the purposes of this analysis, the two parameter choices can thus be considered linked. The four BCRI schemes assessed in this work are summarized in Table 1 and described in the following subsections, listed from least to most absorbing.

## 2.1 Scheme 1: dA1991 ($m_{550nm} = 1.75 - 0.44i$)

The origins of this scheme can be traced to back to the 1970s at least. d'Almeida et al. (1991) tabulated the refractive indices of dust-like, water-soluble, soot, oceanic, sulfate, and mineral aerosol components at wavelengths from 0.300-40 microns, derived using Mie theory. The refractive indices of soot, which is frequently used as interchangeable with BC, were drawn from the tabulation of Shettle and Fenn (1979) which itself compiled data from a number of pre-existing measurements. Following its publication in d'Almeida et al. (1991), this scheme was included in the Optical Properties of Aerosols and

| Abbreviation | Reference | $m_{550*nm}$ | $E(m_{550*nm})$ | $\rho_{BC}$ [g/cm$^3$] |
|---|---|---|---|---|
| dA1991 | d'Almeida et al. (1991) | $1.75 - 0.44i$ | 0.177 | 1.6 |
| BB2006low | Bond and Bergstrom (2006) | $1.75 - 0.63i$ | 0.248 | 1.8 |
| BB2006high | Bond and Bergstrom (2006) | $1.95 - 0.79i$ | 0.255 | 1.8 |
| Besc2016 | Bescond et al. (2016) | $1.48 - 0.84i$ | 0.401 | 1.7 |

**Table 1.** The four BCRI schemes compared in this analysis, listed from least to most absorbing. $m_{550*nm}$: complex refractive index at 550nm. $E(m_{550nm}^*)$: absorption function (Eqn. 1) at 550nm. *For the Besc2016 scheme these values are reported at 532nm, not 550nm.

Clouds database (OPAC; Hess et al., 1998) and entered widespread usage in the climate modelling community, where it is most frequently attributed to one of these two publications. It has since been demonstrated that the dA1991 scheme is inconsistent with observations (Bond and Bergstrom, 2006; Liu et al., 2020); nevertheless, it remains in use in many models (Table 2) and it is the default scheme in CanAM5.1-PAM.

In this work, the spectrally-varying refractive index is drawn directly from the d'Almeida et al. (1991) tabulation. However, we modify the scheme by using a density of 1.6 g/cm$^3$. The original d'Almeida et al. (1991) scheme assumed a density of 1.0 g/cm$^3$ to account for the fact that the particles they measured contained a great deal of air (Hess et al., 1998). This density is far lower than the accepted 1.8 +/- 0.1 g/cm$^3$ (Bond and Bergstrom, 2006), and is an unreasonable value of use in an Earth system model. A density of 1.6 g/cm$^3$ is selected as a compromise.

**2.2 Schemes 2 and 3: BB2006low and BB2006high ($m_{550nm} = 1.75 - 0.63i$ and $m_{550nm} = 1.95 - 0.79i$)**

Bond and Bergstrom (2006) compiled and reviewed laboratory measurements of the optical properties of BC. From these data, they used the RDG approximation and the accepted density of black carbon, 1.8 g/cm$^3$, to propose a range of BCRI values lying along a "void fraction line" which describes BC with varying degrees of air included within its structure. We select $m_{550nm} = 1.75 - 0.63i$ and $m_{550nm} = 1.95 - 0.79i$, the lowest and highest values proposed by Bond and Bergstrom (2006),

for this analysis.

Unlike d'Almeida et al. (1991), Bond and Bergstrom (2006) only provide estimates of the BCRI at 550nm. Flanner et al. (2012) obtained full spectral information for $m_{550nm} = 1.95 - 0.79i$ using the expressions derived by Chang and Charalampopoulos (1990). We use this dataset for our BB2006high scheme and apply an equivalent scaling to the equations of Chang and Charalampopoulos (1990) to obtain the BB2006low spectrum given $m_{550nm} = 1.75 - 0.63i$. For both schemes we assume

a density of 1.8 g/cm$^3$ as used by Bond and Bergstrom (2006).

The BB2006high scheme is used by a number of aerosol models. To our knowledge, no models currently use the BB2006low scheme, although some use intermediate values from Bond and Bergstrom (2006), most commonly $m_{550nm} = 1.85 - 0.71i$ (Table 2). Nevertheless we select $m_{550nm} = 1.75 - 0.63i$ as our intermediate BCRI in order to span the range of estimates from Bond and Bergstrom (2006).

## 2.3 Scheme 4: Besc2016 ($m_{532nm} = 1.48 - 0.84i$)

Bond and Bergstrom (2006) acknowledged in their review that their recommended BCRI could not reproduce the observed mass absorption cross section of black carbon when used in combination with the accepted density of 1.8 g/cm$^3$. Kahnert (2010) investigated their hypothesis that the discrepancy was related to shortcomings of the RDG model used in their calculations, and demonstrated that the choice of optical model was insufficient to explain the underestimation. More recently, Liu et al. (2020) reviewed estimates of the refractive index published since Bond and Bergstrom (2006) in the context of current estimates of the mass absorption cross section and the absorption function $E(m)$ (Section 2.4). Based on this assessment, they recommended refractive indices with $E(m) > 0.32$ in the visible and near infrared, which would rule out the three BCRI schemes described above.

One scheme recommended by Liu et al. (2020) was the Bescond et al. (2016) estimate of $m_{532nm} = 1.48 - 0.84i$, derived from measurements of ethylene flame using a bulk density of 1.74 g/cm$^3$ and a modified version of the RDG approximation which accounts for some internal scattering effects (Yon et al., 2014). This scheme is substantially more absorbing than the previous three at all wavelengths, and to our knowledge has not been used in any Earth system models. The Besc2016 scheme may not be representative of the BC being simulated by Earth system models since most atmospheric BC comes from more complex sources such as coal, propane, or biomass burning, and because BC undergoes rapid morphological transitions after its emission (Ramachandran and Reist, 1995) which alter its optical properties. However, its inclusion in this analysis provides a useful upper bound for the likely impacts of varying the BCRI in Earth system models. Other recent BCRI estimates are discussed in Section 5.

Bescond et al. (2016) report the refractive index at a subset of wavelengths between 266 nm and 1064 nm. Estimates of the refractive index at other wavelengths, which were derived through application of the Kramers–Krönig relation, were obtained though personal communication with the authors and are presented in Li et al. (2024).

## 2.4 Spectral Dependence of the BCRI schemes

The four schemes span a range of absorption, which can be quantified by the wavelength-dependent absorption function $E(m)$ (Bohren and Huffman, 1983):

$$E(m) = \text{Im}\left(\frac{m^2 - 1}{m^2 + 2}\right) \tag{1}$$

Higher values of $E(m)$ indicate an increased tendency for absorption, and the mass absorption cross section of an aerosol is a linear function of $E(m)$ although the details of this relationship depend on aerosol morphology (Liu et al., 2020). At all wavelengths, the dA1991 scheme has the lowest absorption and the Besc2016 the highest. $E(m)$ in the first three schemes increases to both the ultraviolet and infrared, while the Besc2016 scheme decreases slightly to the infrared. All four schemes are fairly constant through the visible and near-infrared (d'Almeida et al., 1991; Chang and Charalampopoulos, 1990; Bescond et al., 2016).

| Aerosol scheme | Host model | References |
|---|---|---|
| dA1991 ($m_{550nm} = 1.75 - 0.44i$) | | |
| AM4.0 | GFDL | Zhao et al. (2018) |
| CLASSIC | ACCESS-ESM1-5, HadGEM2-ES | Bellouin et al. (2011); Ziehn et al. (2020) |
| GOCART | GEOS | Chin et al. (2002); Colarco et al. (2010) |
| SPRINTARS | MIROC-ES2L | Takemura et al. (2002); Hajima et al. (2020) |
| (unnamed) | OsloCTM3 | Myhre et al. (2009) |
| PAM | CanAM5.1-PAM | von Salzen (2006); Cole et al. (2023) |
| BB2006high ($m_{550nm} = 1.95 - 0.79i$) | | |
| ATRAS | CAM5-ATRAS | Matsui (2017) |
| MAM4 | CESM1, E3SM-1-1 | Liu et al. (2012); Wang et al. (2019) |
| MASINGAR | MRI-ESM2-0 | Yukimoto et al. (2019) |
| OsloAero6 | NorESM2-LM | Seland et al. (2020) |
| Alternate recommendation from Bond and Bergstrom (2006) ($m_{550nm} = 1.85 - 0.71i$) | | |
| GLOMAP | UKESM | Sellar et al. (2019) |
| HAM-M7 | ECHAM-HAM | Tegen et al. (2019) |
| SALSA | ECHAM-SALSA | Bergman et al. (2012) |
| TM5-mp3.0 | EC-Earth3-AerChem | van Noije et al. (2021)) |

**Table 2.** Illustrative sample of aerosol schemes that use our selected BCRI; see Section 2 for descriptions of the schemes. To the best of our knowledge no aerosol schemes currently use BB2006low ($m_{550nm} = 1.75 - 0.63i$, the lowest value recommended by Bond and Bergstrom (2006)), or the more recent Besc2016 scheme. However, a number of aerosol models use $m_{550nm} = 1.85 - 0.71i$, an alternate recommendation from Bond and Bergstrom (2006) that falls between our BB2006low and BB2006high BCRI schemes, and we include a selection of those models here.

Although our experiments vary the BCRI at all wavelengths, our analysis predominantly focuses on the characteristics of these schemes at 550nm. This is the wavelength for which Earth system models typically publish aerosol optical data and for which many satellite retrievals are available. In a complementary analysis, Li et al. (2024) assess the optical properties of BC in the dA1991, BB2006high, and Besc2016 schemes, including dependence on wavelength, particle size, and mixing state. The Li et al. (2024) analysis relies on theoretical calculations and a one-dimensional radiative transfer model, while the work presented here explores the impacts of the BCRI on Earth system model simulations.

## 3 Methods

### 3.1 CanAM5.1-PAM

The Canadian Atmospheric Model version 5 (CanAM5; Cole et al., 2023) is the atmospheric component of the Canadian Earth System Model CanESM5 (Swart et al., 2019). Here we use CanAM5.1, which contains a number of technical and process

representation updates as described in Sigmond et al. (2023). Most importantly for aerosol modeling, these updates eliminate the occasional formation of spurious tropospheric dust storms seen in CanESM5.

CanAM5.1 can be run with either of two aerosol schemes: a bulk scheme, which simulates aerosol mass budgets and is used in most applications of the model, or the PLA Aerosol Model (PAM), which we use here. PAM uses the Piecewise Log-normal Approximation (von Salzen, 2006, PLA;) method to simulate aerosol size distributions using a series of truncated, non-overlapping lognormal modes within specified aerosol size sections. Each truncated mode has a specified geometric standard deviation; the magnitudes and mode radii are calculated from the predicted mean masses and number concentrations in each mode at each timestep.

Black carbon and organic carbon are emitted as externally mixed, hydrophobic aerosol, represented by one mode each. Upon ageing (time constant $\tau$=1h during the day and 24h during the night) these tracers are merged with pre-existing internally mixed aerosol, which is represented by 3 modes. Sulfate aerosol can form from ternary homogeneous nucleation of water vapour, gaseous sulfuric acid ($H_2SO_4$), and ammonia, after which point it grows by Brownian coagulation (Tzivion et al., 1987), or by condensation of water vapour, $H_2SO_4$, and secondary organic aerosol precursor gases (von Salzen, 2006; Dunne et al., 2016). All sulfate aerosol is contained within the 3 internally mixed modes and assumed to be fully neutralized by ammonium. Dust and sea salt are externally mixed and are represented by two modes each (Ma et al., 2008; Peng et al., 2012).

Aerosol activation and cloud droplet growth are determined using pre-calculated solutions to the cloud droplet growth equation for an adiabatically rising air parcel near the cloud base (Wang et al., 2022). These solutions are stored in lookup tables and referenced using a numerically efficient iterative approach. The simulated cloud droplet number concentration is used to compute the effective radius of the cloud droplets (Peng and Lohmann, 2003) according to the first indirect effect (Lohmann and Feichter, 2005). In the model configuration used here, the second indirect effect is included via autoconversion of cloud to rain droplets (Cole et al., 2023).

Aerosol sinks in PAM include dry deposition, which depends on ground-surface properties and the near-surface aerosol concentration (Zhang et al., 2001); wet deposition by below-cloud scavenging (Croft et al., 2005); and in-cloud scavenging in convective and layer clouds (Croft et al., 2005; von Salzen et al., 2013). Both below-cloud and in-cloud wet deposition rates are proportional to the precipitation formation rate.

Aerosol optical properties in PAM are determined from pre-computed lookup tables. The tables are generated using Mie theory to determine optical properties as a function of relative humidity, wavelength, and particle size. Although the assumption of spherical particles may be inappropriate for freshly-emitted BC, the majority of the BC simulated by an Earth system model is hours to days old and will be relatively compact and/or internally mixed, making Mie theory a reasonable approximation. For internally mixed aerosol an effective refractive index is computed using the Maxwell-Garnett approximation (Wu et al., 2018), which has been demonstrated to describe the optical properties of coated BC better than volume-weighted or core-shell approximations (Adachi et al., 2010; Stevens and Dastoor, 2019).

Other treatments of aerosol morphology and mixing state would likely yield different absolute values of simulated AAOD and ERFari. However, the focus of this work is on the difference in these quantities between simulations conducted with different BCRI schemes, with other parameterizations held constant. As it is, CanAM5.1-PAM's simulated AAOD is in good

agreement with other Earth System models and with observations, to the extent that the latter can be determined given the associated uncertainties (Figure A1). The AMAP (2021) report found that CanAM5-PAM reproduced the observed vertical profile of BC in both the Arctic and northern midlatitudes particularly well, relative to other models.

In the simulations conducted for this analysis, transient historical sea surface temperatures and sea ice concentrations were specified using the PCMDI observational dataset (Taylor et al., 2000) and historical sea ice thickness for the Northern and Southern Hemispheres were taken from PIOMAS (Zhang and Rothrock, 2003) and ORAP5 (Zuo et al., 2017) reanalyses respectively.

## 3.2 Experimental Design

We simulate four sets of "core ensembles": one control ensemble and one perturbed ensemble for each of the BCRI described above. Each ensemble consists of nine short simulations (2014-2019) and one long simulation (1949-2019). The first year of each is discarded as spinup. In the control ensemble, all emissions of aerosols and greenhouse gases are transient; in the perturbed ensemble, BC emissions are fixed at 1850 levels while other emissions evolve as in the control scenario. ERF is then calculated from the difference in top of atmosphere flux between pairs of control and perturbed runs, following the "ERF_trans" method of Forster et al. (2016). This method of calculating ERF is chosen over the alternative approach in which the control scenario uses preindustrial emissions and the perturbed scenario adds transient emissions of the forcer of interest because the latter method does not account for interactions between species. Finally, the total BC ERF is decomposed into contributions from aerosol-radiation (ERFari), aerosol-cloud (ERFaci), and albedo (ERFalb) interactions following Ghan (2013). In this work we exclusively consider shortwave ERF. Longwave BC ERF is small in CanESM5.1-PAM, consistent with previous findings for models that do not parameterize aerosol impacts on ice and mixed phase clouds (Heyn et al., 2017). Although PAM includes representations of the albedo effects on BC deposited on snow and ice (Namazi et al., 2015) and absorption of solar radiation by BC-containing cloud droplets (Li et al., 2013), in this work we only vary the refractive index of atmospheric BC. ERFaci and ERFalb are thus expected to be similar between the three core ensembles.

The influence of the BCRI on AAOD, ERFari, and tropospheric temperature are quantified by comparing these fields between the four core ensembles. We then compare the changes in ensemble-median AAOD and ERFari that arise from the choice of BCRI to the variation in comparison datasets due to other factors. This comparison is not intended as a comprehensive analysis of BC uncertainties, but rather presents an illustrative sample of relevant uncertainties other than the BCRI.

## 3.3 Comparison Datasets

To contextualize the sensitivity of AAOD and ERFari to variations in BCRI, we consider three different comparison datasets characterizing aspects of the uncertainties in these quantities.

Our first comparison investigates the impact that recent updates to aerosol emission inventories have on simulated AAOD. In the core ensembles, anthropogenic aerosol emissions are taken from the Community Emissions Data System (CEDS) 2021-04-21 release (O'Rourke et al., 2021). CEDSv2021 emissions not only extend the historical emissions used in CMIP6 to more recent years but also include several back-corrections, most notably reducing the emissions of BC, organic carbon, and

sulfur dioxide over China (O'Rourke et al., 2021; Wang et al., 2021). Biomass burning emissions in the core ensembles are

taken from the CMIP6 historical inventory for 1950-2014 and the Global Fire Emissions Database (GFED v4.1s; van der Werf et al., 2017) for 2015-2019. To investigate the impact of these selections, we run a single simulation forced with the more commonly used CMIP6 historical and SSP2-4.5 anthropogenic and biomass burning emissions, otherwise identical to the low-absorption core ensemble. We compare the AAOD from this simulation against each realization of the low-absorption ensemble in turn, yielding a 9-member ensemble of differences. We emphasize that this comparison does not represent the total uncertainty in either anthropogenic or biomass-burning aerosol emissions, as we are comparing two versions of the same anthropogenic emissions inventory and extending the biomass burning emissions with the same observational product as was used in the creation of the CMIP6 historical and SSP inventories. Instead this comparison shows the sensitivity of AAOD to recent improvements in both sets of emissions.

Our second comparison considers observational uncertainty. We compare estimates of AAOD from the Multi-angle Imaging SpectroRadiometer (MISR; Diner et al., 1998) and the Polarization and Directionality of the Earth's Reflectances instrument with the Generalized Retrieval of Atmosphere and Surface Properties algorithm (POLDER-GRASP; Dubovik et al., 2011), selected for their availability in Level 3 gridded format. MISR and POLDER-GRASP bracket the range of AAOD simulated by current Earth system models (Figure A1). We consider satellite observations rather than ground-based or in-situ measurements due to the challenge in comparing point-source data against gridded model results (Wang et al., 2016, 2018; Schutgens et al., 2021). POLDER-GRASP data are not available for the 2015-2019 study period, so we instead compare these satellites over the 5-year period 2007-2011. The difference between MISR results for 2015-2019 and 2007-2011 is substantially smaller than the difference between MISR and POLDER-GRASP data for 2007-2011, indicating that use of these alternate years is unlikely to affect our conclusions. This comparison is not intended to be a detailed evaluation of the uncertainty in remotely-sensed AAOD; such assessments can be found in, for example, Schutgens et al. (2021). Instead it provides an estimate of the range of AAOD that can be obtained from different instruments. As such it does not account for differences in sampling between the two satellites or between the observed and simulated AAOD fields.

Our final comparison considers the range of AAOD and ERFari reported in recent multimodel assessments from the literature. This comparison folds in many sources of model uncertainty, including differences in the treatment of mixing state which has been shown to have substantial impact on simulated BC absorption (Sand et al., 2021; Gliß et al., 2021), as well as differences in the parameterization of aerosol transport and deposition. The individual assessments are described in Section 4.

## 4 Results

### 4.1 Absorption Aerosol Optical Depth

Modifying the BCRI directly modifies BC absorption, and we thus begin by assessing the its impact on AAOD (Figure 1). Increasing BC absorption from the dA1991 scheme to the BB2006low scheme increases global-mean 2015-2019 AAOD by 27%; increasing from dA1991 to BB2006high increases AAOD by 42 %; and increasing from dA1991 to Besc2016 increases AAOD by 59%. Absolute increases are, unsurprisingly, largest over major source regions, but the medium- and high-absorption

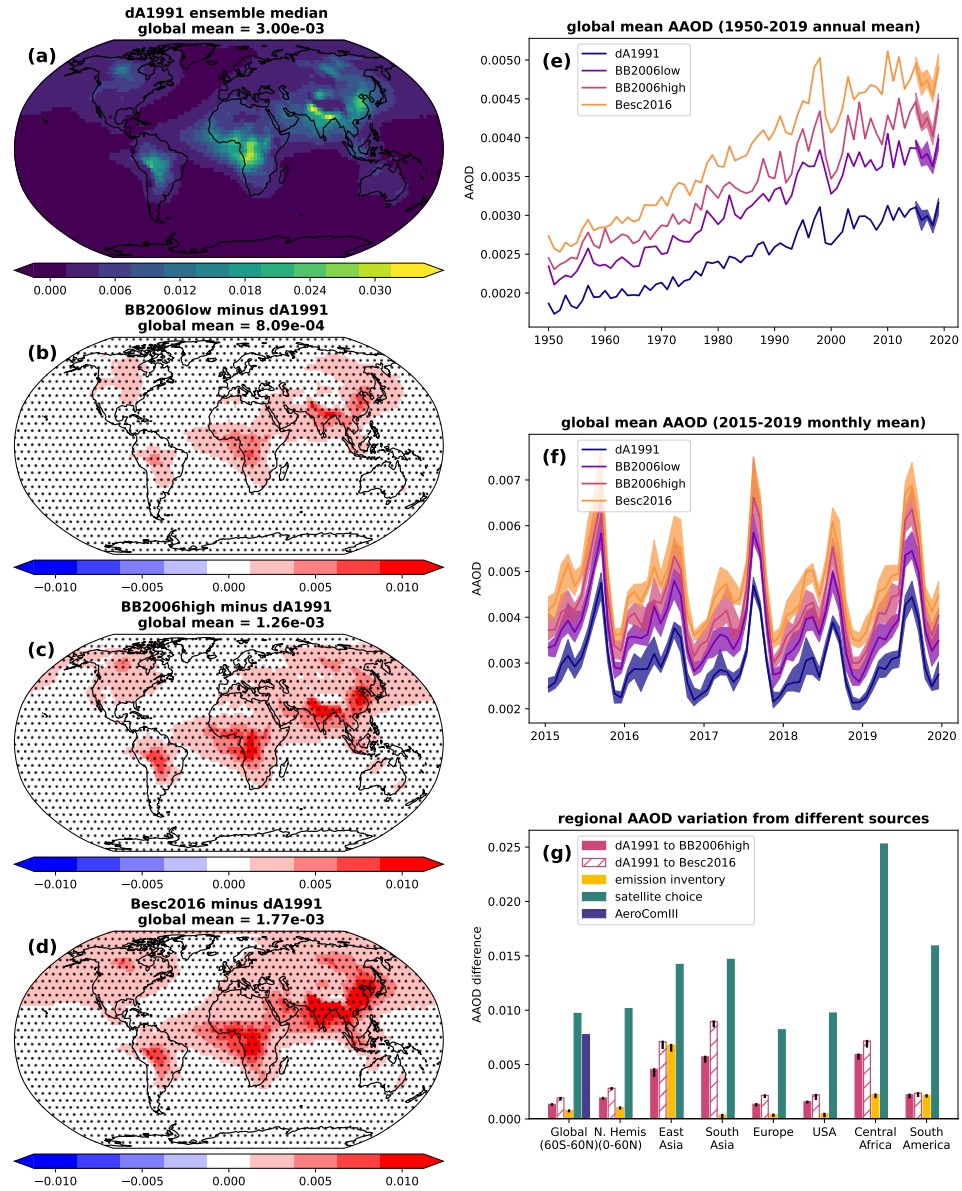

**Figure 1.** Maps show ensemble-median, 2015-2019 mean AAOD in the **(a)** low-absorption dA1991 ensemble, **(b)** BB2006low minus dA1991 ensemble, **(c)** BB2006high minus dA1991 ensemble, and **(d)** Besc2016 minus dA1991 ensemble. Stippling in **(b)**, **(c)**, and **(d)** indicates regions where the difference between time-averaged ensembles is statistically significant at the 5% level according to a two-sided Students t-test; the global-mean AAOD differences are statistically significant in all three cases. Timeseries show global-mean AAOD in the four BCRI ensembles over **(e)** 1950-2019 and **(f)** 2015-2019. Panel **(g)** compares the difference in ensemble-median, 2015-2019 mean AAOD between low and high absorption ensembles (dark pink; dA1991 to BB2006high in solid colour, dA1991 to Besc2016 hatched) to the spread in AAOD obtained from simulations using different emission inventories (gold) and to the range in remotely-sensed AAOD from different satellites (teal) in 8 different geographic regions, and to the overall range in AAOD from AeroCom Phase III models (Sand et al., 2021) for the near-global region only (indigo). Shaded envelopes in panels **(e)** and **(f)**, and error bars in panel **(g)**, denote the 5th-95th percentile range across ensembles.

ensembles are statistically significantly different from the low-absorption ensemble everywhere (Figure 1b, c, d). The four ensembles of global-mean AAOD are clearly separated, with no overlap between annual-mean AAOD (Figure 1e) and little overlap between monthly-mean AAOD (Figure 1f). As absorption increases, so too does the magnitude of the trend in AAOD over 1950-2019 (Figure 1e).

Figure 1g compares the regional increases in AAOD from varying the BCRI to the increases obtained by varying the aerosol emissions and to the differences in observed AAOD from the MISR and POLDER-GRASP satellites. Two BCRI-induced changes are shown: solid bars give the AAOD difference between dA1991 and BB2006high ensembles, and hashed bars the difference between dA1991 and Besc2016. Region definitions are provided in Appendix B. The "near-global" and "Northern Hemisphere" domains exclude latitudes poleward of $60°$ where the satellite retrievals are poorly sampled. Most of the regional variation in the increase of AAOD caused by varying the BCRI comes from differences in the local BC burden (Figure B1); the relative increase is fairly consistent, varying from 39-47% when comparing BB2006high to dA1991.

The choice of aerosol emission inventory has the greatest impact in regions where the two inventories are the most different and where the baseline emissions are high. The largest emissions-based AAOD increase, both in absolute terms and relative to the BCRI-induced increase, occurs in East Asia where both conditions are satisfied; the smallest occurs in South Asia where the two inventories are nearly identical. These regional increases are a factor of 1.5 larger than, and a factor of 16 smaller than, the AAOD differences between dA1991 and BB2006high ensembles respectively. The $60°$S-$60°$N average change resulting from the updated emissions is approximately 60% of the BCRI increase. Note that these values do not indicate the overall uncertainty in global or local emissions, only the spatial distribution of updates to the inventories being considered. For example, Pan et al. (2020) report a factor of seven difference in Southeast Asian biomass burning emissions from different inventories (a more accurate representation of emission uncertainties in this region), whereas we see almost no change between the inventory versions considered in this assessment.

In all regions the range of AAOD between different satellites is substantially larger than the range of simulated AAOD in CanAM5.1-PAM under different configurations. This is expected given the challenges in constraining remotely sensed AAOD (Samset et al., 2018b). Even so, in regions where the BC burden is high the impact of BCRI on AAOD can be as much as two thirds as large as the difference between satellites. Averaged over the near-global domain, the inter-satellite discrepancy is a factor of five larger than the AAOD difference between dA1991 and Besc2016. The satellite differences are expected to be largest in regions where characteristics of the geography increase retrieval uncertainty (e.g. regions with more reflective surfaces) and in regions where the satellites differ in their sensitivity to the dominant aerosol type.

Finally, for the near-global domain only we compare with the spread in global-mean AAOD from AeroCom Phase III models (Sand et al., 2021). AeroCom Phase III estimates of the global-mean AAOD in simulations forced with 2010 emissions range from $2.04 \times 10^{-3}$ to $9.78 \times 10^{-3}$ for an overall range of $7.75 \times 10^{-3}$ or approximately 6 times that obtained by varying the BCRI alone. This is similar to the range of AAOD between satellites, and as shown in Figure A1, the distribution of zonal-mean AeroCom Phase III AAOD is approximately bracketed by the zonal-mean AAOD from MISR and POLDER-GRASP. The 15 AeroCom models have BCRI spanning between our dA1991 and BB2006high schemes, but BCRI alone does not explain the intermodel spread, as discussed further in Section 5.

Overall, the sensitivity of simulated AAOD to the choice of BCRI is comparable to its sensitivity to recent updates to the aerosol emission inventory within the same model. Between models, or between satellites, the global-mean AAOD spread is a factor of 5 to 7 larger.

Despite the fact that BC generally makes up a small fraction of total aerosol extinction (Bond et al., 2013), we do see regionally significant increases in total aerosol optical depth (AOD) when BC absorption is increased (Appendix C). These changes are particularly evident in central and southern Africa, South America, and the northern latitudes. These are all regions with high emissions from biomass burning, which are associated with a higher ratio of BC emissions. Heavily polluted regions, such as South and East Asia, do not show an AOD dependence on BCRI, likely because AOD in these regions is dominated by sulfate. There is not a statistically significant difference between globally-averaged AOD in the four core ensembles.

## 4.2 Aerosol-Radiation Forcing

The BCRI directly affects the interaction of BC with incoming solar radiation and therefore the aerosol-radiation forcing ERFari. Figure 2 summarizes the variation in shortwave BC ERFari across our four ensembles.

Increasing BC absorption from the dA1991 scheme to the BB2006low scheme increases global-mean 2015-2019 BC ERFari by 32%; increasing from dA1991 to BB2006high increases BC ERFari by 47%; and increasing from dA1991 to Besc2016 increases BC ERFari by 100%. These increases are largely confined to the Northern Hemisphere, and for the BB2006high and Besc2016 ensembles, are statistically significant in most regions where the dA1991 BC ERFari is significantly nonzero (Figure 2c,d). Global-mean BC ERFari in the BB2006low, BB2006high, and Besc2016 ensembles are significantly different from the dA1991 ensemble at the 5% level. There is more interannual variability in the timeseries of global-mean BC ERFari than there is in AAOD, likely due to the dependence of ERFari on cloud fields (Figure 2e,f). Despite this variability, ERFari trends over 1950-2019 increase with BCRI as the AAOD trends do.

The uncertainty in BC ERFari attributable to the choice of BCRI can be compared to results from two recent literature assessments. We first compare with the multimodel range of BC ERFari assessed by Thornhill et al. (2021), and then with the statistical uncertainty in BC ERFari reported by the Artic Monitoring and Assessment Program (AMAP) 2021 report. We emphasize that these are fundamentally different quantities with different interpretations. Furthermore, Thornhill et al. (2021) and AMAP (2021) estimate ERF for different time periods, assuming different emission inventories, and using different methodology in their calculations, so the studies' best estimates are not directly comparable to our results or to each other. The comparisons are nevertheless useful in contextualizing how much of an impact uncertainty in the BCRI has on that of BC ERFari.

Thornhill et al. (2021) calculated ERF for numerous aerosol and greenhouse gas species for 1850-2014 based on results from AerChemMIP (Collins et al., 2017). Eight models provided estimates of total BC ERF but only four decomposed this ERF into contributions from radiation, cloud, and surface albedo forcings. These four models (CNRM-ESM2, MRI-ESM2, NorESM2, and UKESM1) simulated BC ERFari of 0.37 W/m$^2$, 0.13 W/m$^2$, 0.35 W/m$^2$, and 0.26 W/m$^2$ respectively, for an overall range of 0.24 W/m$^2$. This is a factor of 1.3 (2.6) larger than the difference in ERFari simulated by our dA1991 and Besc2016 (BB2006high) ensembles. The four models use a narrow range of high-absorption BCRI: $m_{550nm} = 1.95 - 0.79i$

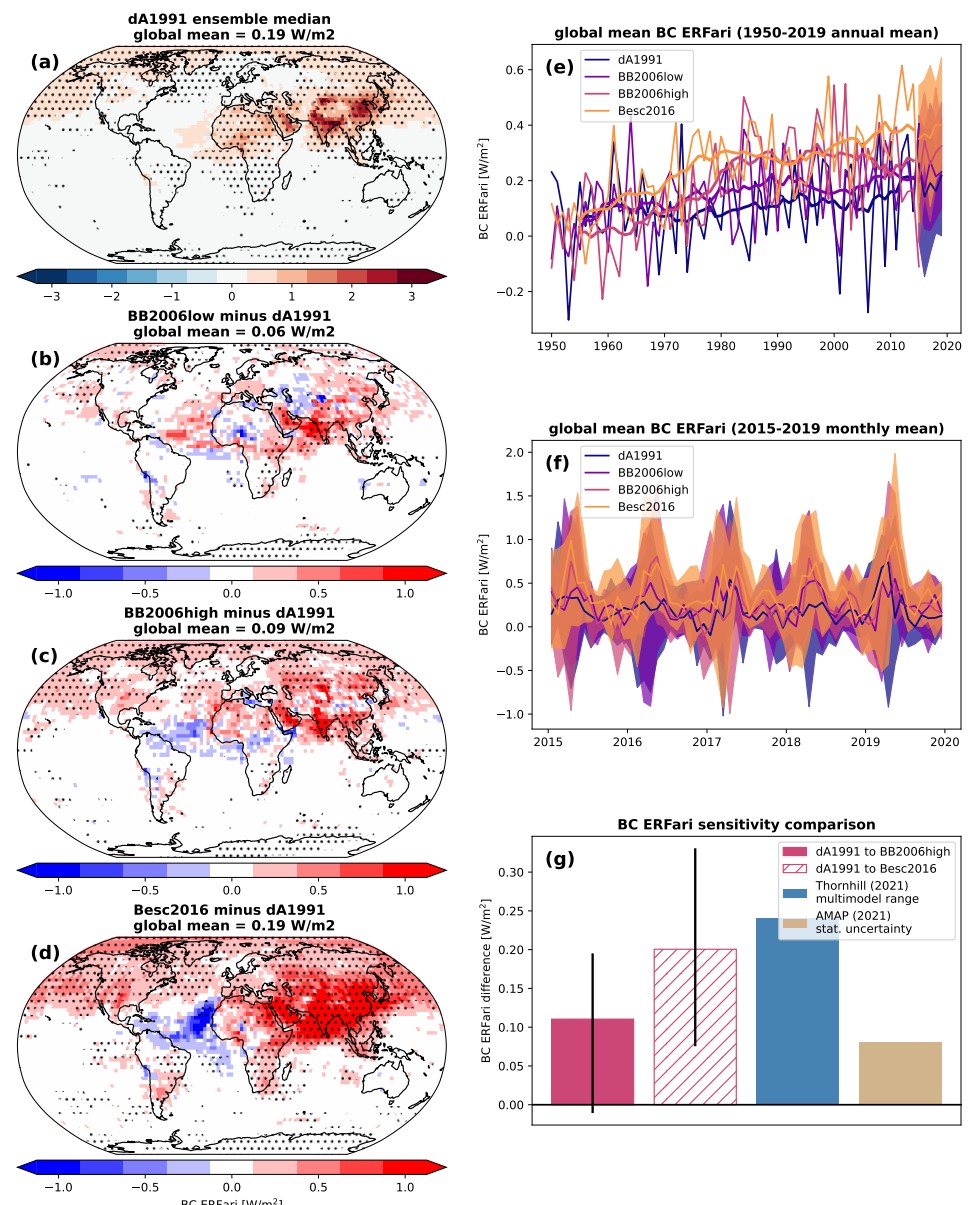

**Figure 2.** Panels **(a-f)**: as Figure 1 but for BC ERFari. Heavy lines in **(e)** show 11-year rolling means. Panel **(g)** compares the difference in BC ERFari between low and high absorption ensembles (dA1991 to BB2006high in solid colour, dA1991 to Besc2016 hatched) to the multimodel range from the Thornhill et al. (2021) assessment (blue), and to the statistical uncertainty in the AMAP (2021) assessment (tan). The global-mean increase of 0.09 W/m$^2$ referenced in the title of panel **c** does not match the median increase of 0.11 W/m$^2$ shown in panel **f** because panel **c** shows the difference between ensemble medians, whereas panel **f** shows the ensemble distribution of global-mean differences between individual realizations.

(MRI-ESM2 and NorESM2), $m_{550nm} = 1.85 - 0.71i$ (UKESM), and $m_{550nm} = 1.83 - 0.74i$ (CNRM-ESM2). The assessed range thus did not stem primarily from differences in the BCRI, although differences in the treatment of mixing states could still result in different levels of BC absorption. The results of Thornhill et al. (2021) provide the basis for the assessed BC ERFari in the latest Intergovernmental Panel on Climate Change (IPCC) assessment report, AR6 (Szopa et al., 2021); as AR6

did not itself quote an uncertainty range for BC ERFari, we do not otherwise include it in this comparison.

AMAP (2021) assessed radiative forcing for 1850-2015 using a different aerosol emission inventory than Thornhill et al. (2021). Five models contributed data for BC ERF, but only two provided the decomposition into ERFari: MRI-ESM2 and CanAM5-PAM, with BCRI of $m_{550nm} = 1.95 - 0.79i$ and $m_{550nm} = 1.75 - 0.44i$ respectively (equivalent to the BB2006high and dA1991 schemes assessed here). The reported ERFari derived from these two models was $0.27 \pm 0.04$ W/m$^2$. Unlike

the Thornhill et al. (2021) and AeroCom results discussed above, this $\pm 0.04$ W/m$^2$ is an estimate of statistical uncertainty rather than a multimodel range. It represents the precision to which ERFari can be determined when derived from 100-year integrations of two Earth system models. The overall uncertainty range of 0.08 W/m$^2$ is $\sim$80% of the spread we obtain by varying the BCRI across the same range within one model, suggesting that a different choice of BCRI in either model could have materially impacted the assessed ERFari.

Varying the BCRI, and thus the absorption, of atmospheric BC is found not to have a statistically significant impact on the aerosol-cloud forcing in this experiment. We did not vary the BCRI within cloud droplets, so the only impact on clouds would be via the impact of changes in atmospheric temperature profiles, discussed below. These changes are found to be small relative to the variability of simulated cloud fields. Similarly, the radiative forcing from albedo changes was not found to vary with BCRI scheme because we did not vary the refractive index of BC deposited on snow and ice. The only change in the total

BC ERF was thus from the aerosol-radiation component. For the BB2006low and BB2006high schemes, this change was too small to result in a statistically significant increase in total ERF. However, the Besc2016 scheme led to a statistically significant increase in global-mean BC ERF relative to the dA1991 scheme, from -0.02 W/m2 to +0.24 W/m2.

## 4.3 Temperature and Precipitation

Black carbon influences global and regional temperature (AMAP, 2021; Bond et al., 2013; von Salzen et al., 2022), mean and
350 extreme precipitation (Samset et al., 2016; Samset, 2022; Sand et al., 2020), and the Asian monsoon (Ganguly et al., 2012; Li et al., 2016; Westervelt et al., 2020; Xie et al., 2020). The sensitivity of these fields to the BCRI cannot be assessed in this study as the four ensembles used the same set of prescribed sea surface temperatures. We can, however, examine temperature changes from the radiative heating of the atmosphere by BC in the mid and upper troposphere, and draw on other works to estimate potential precipitation changes.

Figure 3 illustrates the vertical distribution of temperature changes obtained by varying the BCRI. Although near-surface air temperatures are constrained by the prescribed sea surface temperatures, statistically significant zonal-mean warming is evident over the northern midlatitudes starting at about 850 hPa. At 600 hPa and above, this warming is mostly confined to regions above and downwind of East and South Asia (not shown); at lower altitudes, significant warming is also apparent over or downwind of Africa and South America. Large temperature responses are apparent at the poles in Figure 3a,b but note

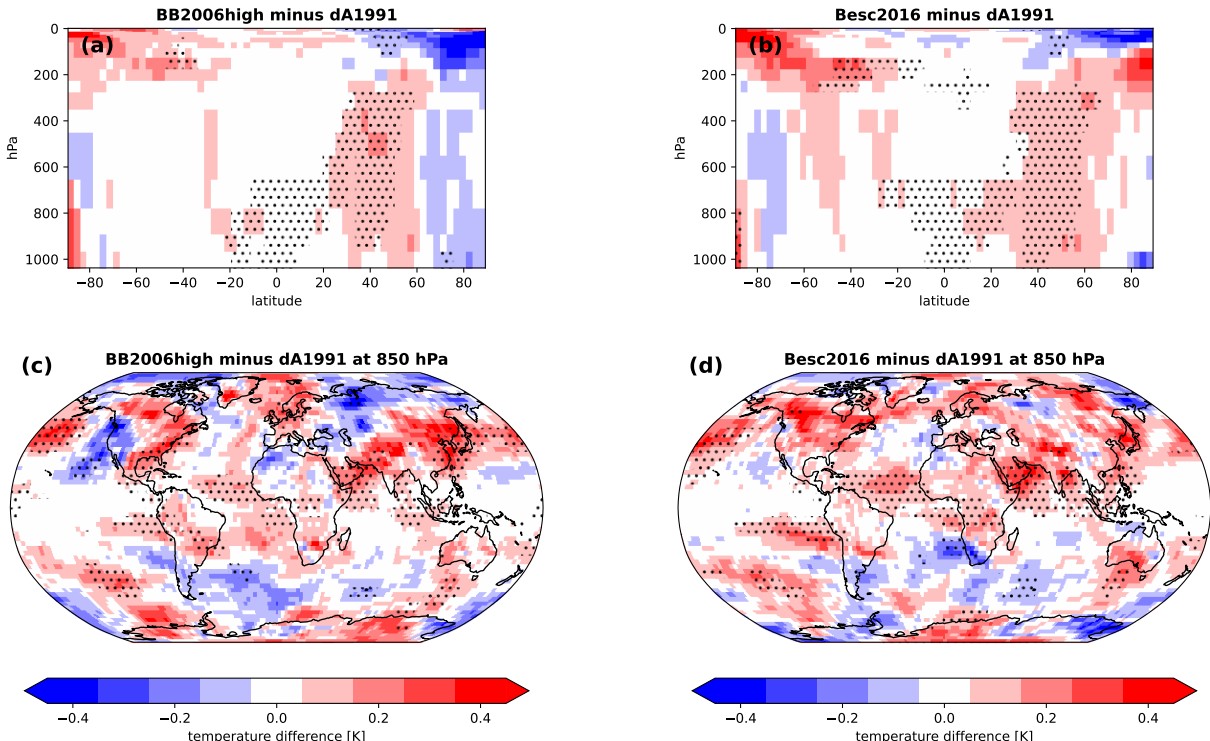

**Figure 3.** Atmospheric temperature differences obtained by increasing the absorption of BC from that in the dA1991 scheme to that in the BB2006high scheme (left) or the Besc2016 scheme (right). **(a,b)** Vertical profile of zonal-mean temperature differences. **(c,d)** Spatial distribution of temperature differences at 850 hPa. Stippling indicates statistically significant temperature anomalies.

that these changes are not statistically significant, and that they correspond to small geographic areas. The mid-tropospheric warming does not appear sufficient to impact local cloudiness in our model: there are not statistically significant differences between the low- and high-absorption cloud fractions at these levels.

To estimate the potential impact of BCRI on precipitation, we draw on the results of Samset (2022) who derived an expression for global-mean precipitation suppression as a function of changes in AAOD:

$$\Delta P = \frac{dP}{dAbs} \times \frac{dAbs}{dAAOD}\, \Delta AAOD = -4646 \pm 1600\, \mathrm{mm\, y}^{-1}(\mathrm{unit\, AAOD})^{-1} \qquad (2)$$

where $P$ denotes the precipitation rate in mm/year and $Abs$ denotes atmospheric absorption, defined as the difference between top of atmosphere and surface radiative forcings in units of $\mathrm{Wm}^{-2}$. A numerical estimate of the first factor in Equation 2 was derived from historical simulations in AeroCom Phase II and CMIP6 models, and the second factor was drawn from the results of Samset et al. (2016) and Persad et al. (2022). Applying Equation 2 to the global-mean, ensemble-median increase in AAOD obtained by changing from the dA1991 to BB2006high scheme yields an estimated precipitation suppression of 5.9 $\pm$

2.0 mm/year. It is extremely unlikely that this signal would be detectable in the global mean in our simulations. It is possible, however, that regional changes could be considerably larger.

## 5  Discussion

We have demonstrated that increasing the BCRI across the range of values commonly used in the climate modeling literature
can increase global-mean AAOD by 42% and BC ERFari by 47%, and using a more recent estimate of the BCRI can increase global-mean AAOD by 59% and BC ERFari by 100%. The resulting increase in the absorption of solar radiation can increase temperatures in the mid and upper troposphere by as much as 0.4°C over major BC source regions, even without considering the potential impacts of BC on sea surface temperatures and sea ice which we have not addressed. For these key BC-relevant fields, therefore, the choice of BCRI is an important and perhaps underappreciated one.

In order to motivate their review on constraining global and regional aerosol absorption, Samset et al. (2018a) briefly explored the effects of modifying the optical properties of BC in CESM1.2. They modified the BCRI by an amount sufficient to increase the resulting AAOD by approximately one standard deviation of the reported AAOD range from AeroCom Phase II (Myhre et al., 2009). Although they do not report the change in BCRI that was necessary to obtain this increase, the result was an increase in AAOD from $3.0 \times 10^{-3}$ to $4.3 \times 10^{-3}$ ($+1.3 \times 10^{-3}$ or 43%), almost identical to the difference between our dA1991 and BB2006high ensembles. This increase in absorption led to a global-mean instantaneous ERF of 0.2 W/m$^2$ relative to the control configuration. In our analysis the median increase in total BC ERF between these BCRI schemes was 0.1 W/m$^2$, but this increase was not statistically different from zero.

Diversity in simulated aerosol absorption in AeroCom Phase III models was investigated by Sand et al. (2021). As referenced in Section 4.1, AeroCom Phase III models exhibited a wide range of absorption, with global-mean AAOD attributable to BC ranging from $0.7 \times 10^{-3}$ to $7.7 \times 10^{-3}$. However, Sand et al. (2021) demonstrated that the models did not display a clear relationship between BCRI and overall absorption, because aerosol absorption is not purely a function of BCRI but rather the result of many competing and uncertain processes (Gliß et al., 2021; Koch et al., 2009; Myhre et al., 2009; Sand et al., 2021). The authors attribute the AeroCom Phase III spread to three main factors: diversity in the simulated mass load, which ranged from 0.13 to 0.51 mg/m$^2$, driven by differences in deposition processes and thus in aerosol lifetime; diversity in the prescribed density of black carbon, which ranged from 1.0 to 2.3 g/cm$^3$; and diversity in the BCRI, which ranged from 1.75-0.44i to 1.95-0.79i. These three factors contributed similarly to the overall model spread. There was also substantial diversity in the treatment of mixing state, both in terms of the mixing itself and in the calculation of resultant effective optical properties of the mixture, between the participating models. Between the diversity in effective refractive index due to differences in absorption enhancement from mixing, and the diversity of assumed BC density, the correlation between BCRI and mass absorption cross section was found to be low (0.2). Thus while BCRI is an influential parameter choice when all else is held equal, its impact is modulated by the competing effects of other parameterizations.

We have assessed four BCRI schemes here, but many others exist. As well as the Bescond et al. (2016) scheme assessed here, the review by Liu et al. (2020) highlighted the Williams et al. (2007) values of $m_{635nm} = 1.75 - 1.03i$ and $E(m_{635nm}) = 0.365$

as being consistent with current estimates of the absorption function. Williams et al. (2007) reported measurements at 635nm and 1310nm and did not extrapolate to other wavelengths, but assuming a relatively flat $E(m)$ through the visible range, this would indicate a degree of absorption somewhere between our BB2006high and Besc2016 schemes. Other estimates which have been widely used in the combustion science literature, such as the Janzen (1979) value of $m = 2.0 - 1.0i$ for all visible wavelengths or the more recent Moteki et al. (2010) $m_{1064nm} = (2.26 \pm 0.13) - (1.26 \pm 0.13)i$, also yield $E(m)$ between BB2006high and Besc2016, but with values low enough that they are not recommended by Liu et al. (2020).

Refractive indices determined from laboratory measurements may not be representative of atmospheric black carbon. For instance, BC generated by a simple, clean laboratory flame will likely have a different temperature history – and thus, different optical and structural properties – than that generated by the more complex sources responsible for most atmospheric BC (Bond and Bergstrom, 2006). Combustion experiments also measure freshly emitted particles, which may have substantial morphological differences from hours-to-days old atmospheric BC. The recent work by Moteki et al. (2023), who measured the refractive indices of atmospheric BC particles sampled during a scientific cruise in the northwest Pacific may be more suitable for use in climate models. By combining their optical measurements with the constraints imposed by the accepted mass absorption cross section of BC, they obtained a range of plausible refractive indices suitable for describing atmospheric BC. Their recommended value, $m_{633nm} = 1.95 - 0.96i$ with $E(m_{633nm}) = 0.297$, also falls between the BB2006high and Besc2016 schemes.

## 6   Conclusions

The radiative forcing of black carbon is subject to many complex, interconnected sources of uncertainty. We have isolated one key factor, the refractive index (BCRI), and demonstrated its impact on the simulation of several fields. With all other parameterizations held equal, increasing the BCRI across the range of values commonly used in Earth system models, ($m_{550nm} = 1.75 - 0.44i$ to $m_{550nm} = 1.95 - 0.79i$) can increase global-mean AAOD by 42% and BC ERFari by 47%. A more recent laboratory estimate, $m_{532nm} = 1.48 - 0.84i$, serves as a likely upper limit to the absorption of atmospheric BC; it yields AAOD and BC ERFari increases of 59% and 100% respectively, relative to the low-absorption case. The impacts of varying BCRI on AAOD are comparable to the effects of recent updates to anthropogenic and biomass-burning aerosol emission inventories, and in BC source regions, the difference in AAOD between low- and high-absorption ensembles is up to two thirds as large as the difference in AAOD retrieved from MISR and POLDER-GRASP satellites. The increase in BC ERFari is comparable to the uncertainty in recent literature estimates. While we do not attribute the spread in previous model estimates of AAOD and ERFari to diversity in BCRI – rather, the multimodel spread arises from a combination of BC parameter choices including BCRI and density, the treatment of mixing and ageing, and the parameterization of transport and deposition processes, among other factors – the similar magnitude emphasizes the importance of considering BCRI choices in model development and multimodel comparisons.

*Code and data availability.* The full CanESM5 source code is publicly available at https://gitlab.com/cccma/canesm/ (last access: 22 Jan 2025). Simulation data used in this project are available at https://crd-data-donnees-rdc.ec.gc.ca/CCCMA/publications/2025_Digby_black_carbon_refractive_index/ (last access 22 Jan 2025). Figures were created using Matplotlib version 3.7.1 (Hunter, 2007; Caswell et al., 2023), available under the Matplotlib license at https://matplotlib.org/. The analysis scripts are available via Zenodo (Digby, 2025).

Several datasets were used to develop the aerosol emission inventories in this work. Anthropogenic emissions for the core ensembles were taken from the CEDS v2021-04-21 data release (O'Rourke et al., 2021). Biomass burning emissions for 2015-2019 for the core ensembles were taken from GFEDv4.1s, described in van der Werf et al. (2017) and available at https://www.geo.vu.nl/~gwerf/GFED/GFED4/. CMIP6 anthropogenic and biomass burning emissions are available from the Earth System Grid Federation (ESGF) at https://aims2.llnl.gov/search/input4mips/.

MISR satellite observations are available via the NASA Langley Atmospheric Science Data Center (NASA/LARC/SD/ASDC, 2008). POLDER-GRASP observations are available from the POLDER Data Release site (POL, 2015); the "compnents" product was used in this analysis. Level 3 monthly data at 1x1 degree resolution were used for both datasets.

## Appendix A: CanAM5.1-PAM Evaluation

The AAOD simulated by CanAM5.1-PAM is compared with that from AerCom Phase III models (Sand et al., 2021) and satellite retrievals in Figure A1.

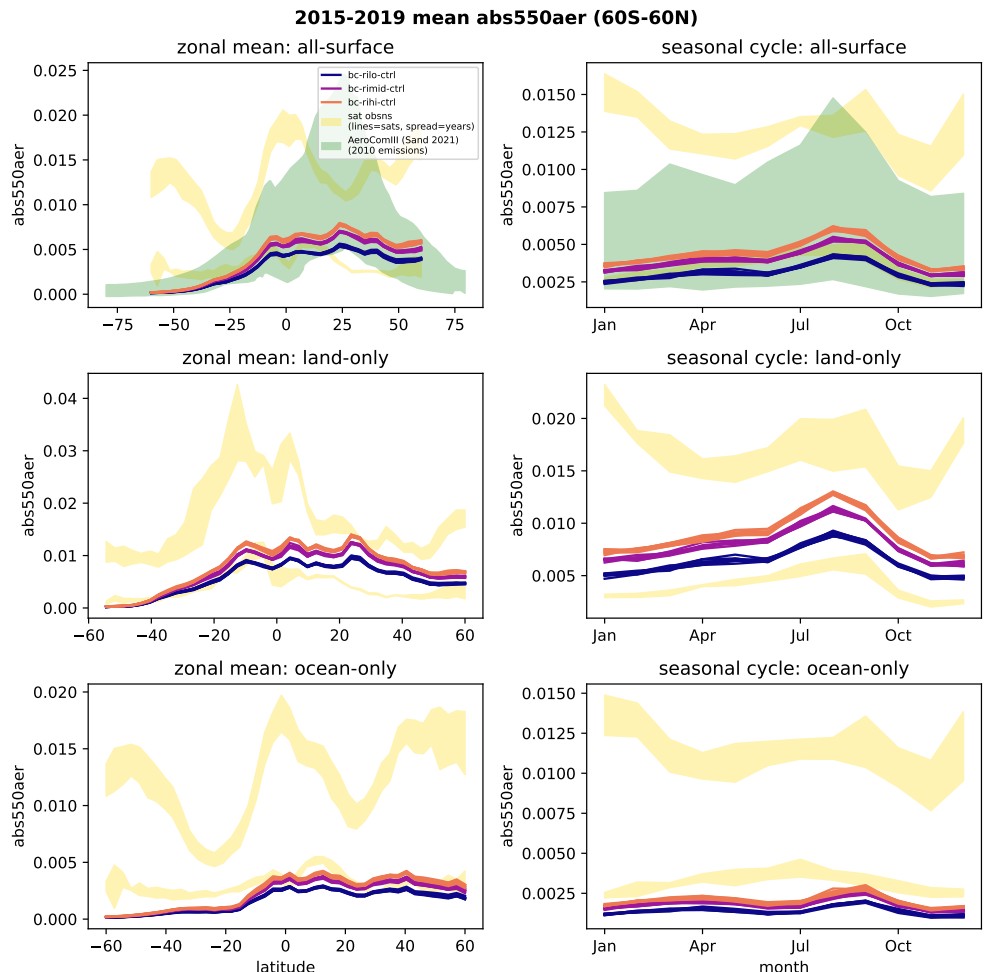

**Figure A1.** 2015-2019 zonal mean (left column) and seasonal cycle (right column) of AAOD in PAM ensembles (coloured lines), satellite retrievals (yellow envelopes), and AerCom Phase III models as reported in Figure 2 of Sand et al. (2021) (pink envelope). For PAM models, individual lines denote individual realizations, averaged over 2015-2019. For satellites, the width of the envelope shows the min-max range over the 5 years in question (2015-2019 for MISR, the lower envelope, and 2006-2010 for POLDER-GRASP, the higher envelope). The AeroComII envelope indicates the full min-max range across models, for simulations forced with 2010 emissions. PAM is generally in closer agreement with MISR than POLDER-GRASP, and within the range of both zonal means and seasonal cycles simulated by AeroCom Phase III models.

## 450 Appendix B: Region Definitions

The regions used in Figures 1 and C1 are defined in Table B1. Figure B1 shows these region boundaries overlaid on a map of black carbon burden from the low-BCRI ensemble.

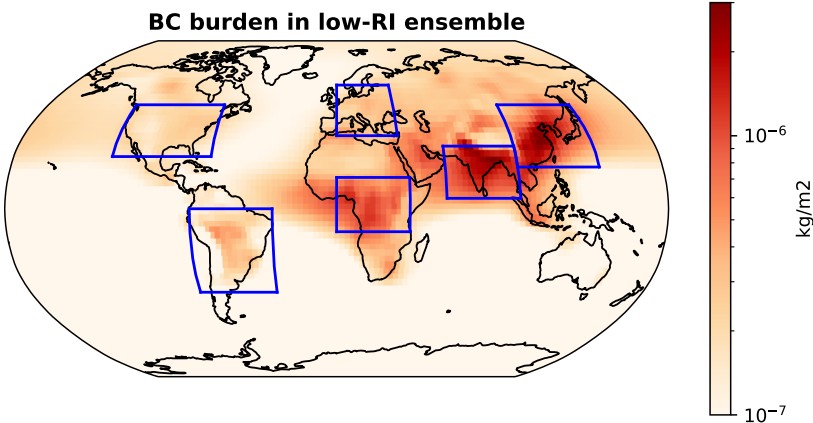

**Figure B1.** Black carbon burden (2015-2019 mean, ensemble median) in the low-BCRI ensemble, with analysis regions indicated in blue. Not shown are the near-global (60°S-60°N) and northern hemisphere (0°N-60°N) domains. Region definitions are listed in Table B1.

| Region | longitude range [°E] | latitude range [°N] |
|---|---|---|
| Near-Global | 0, 360 | -60, 60 |
| Northern Hemisphere | 0, 360 | 0, 60 |
| East Asia | 100, 145 | 20, 50 |
| South Asia | 60, 100 | 5, 30 |
| Europe | 0, 35 | 35, 60 |
| USA | 235, 290 | 25, 50 |
| C. Africa | 0, 40 | -11, 15 |
| S. America | 280, 325 | -40, 0 |

**Table B1.** Definitions of the regions used in Figures 1 and C1 and shown in Figure B1. The East Asia and South Asia definitions are taken from the SREX regions used in IPCC AR5, and the C. Africa region combines the SREX regions EAF and WAF (excluding the portion of WAF west of the prime meridian).

**Appendix C: Total-AOD Results**

Figure C1 reproduces Figure 1 but for total AOD.

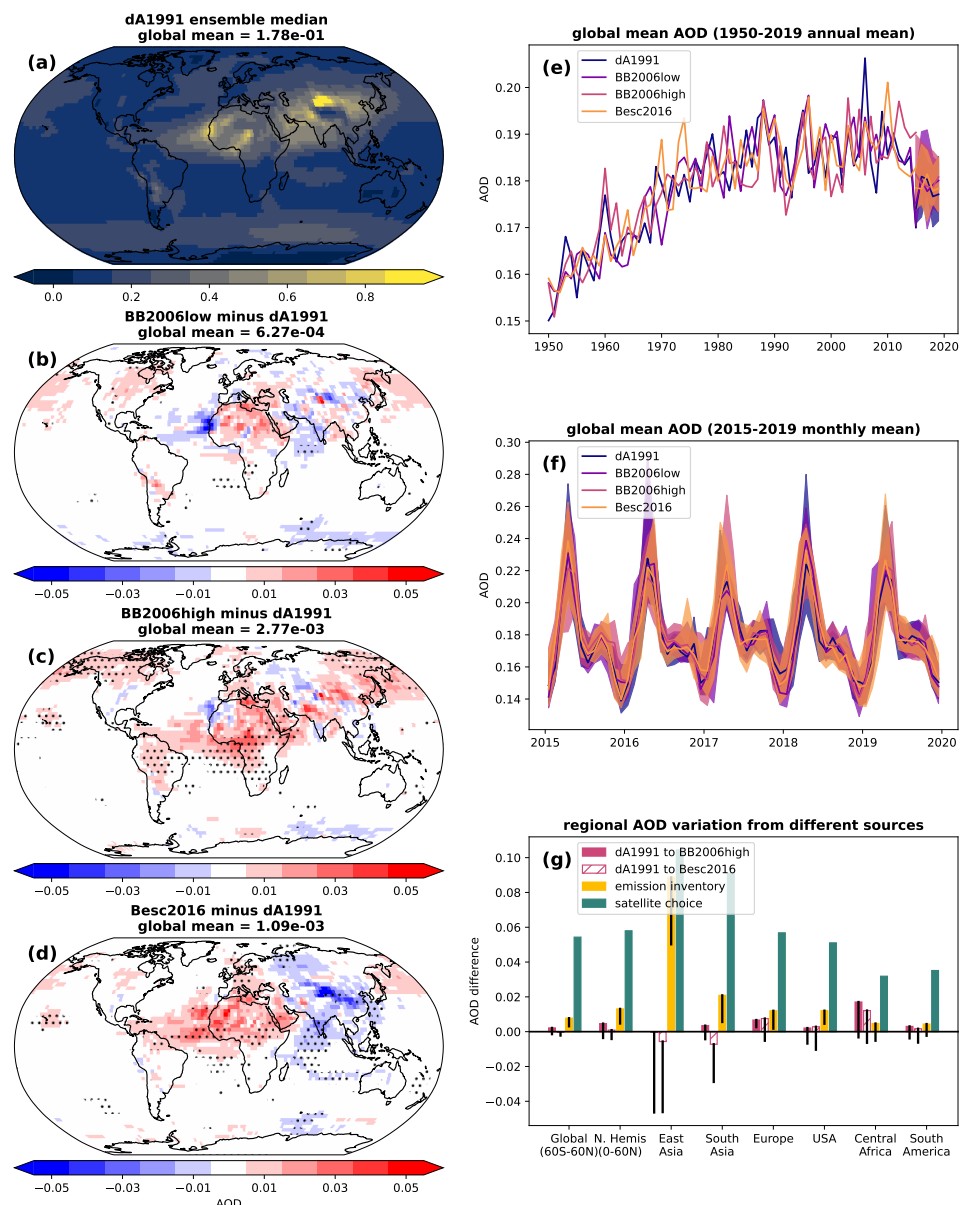

**Figure C1.** As Figures 1 and 2 but for total aerosol optical depth. BCRI has regionally significant impacts on AOD but the global mean does not differ significantly between low- and high-absorption ensembles. In panel (g), AOD retrievals are taken from MISR, the Moderate Resolution Imaging Spectroradiometer (MODIS; Platnick et al., 2017; King et al., 2013), comparing Aqua and Terra results separately), and the Cloud-Aerosol Lidar with Orthogonal Polarization (CALIOP, Winker et al. (2009)). The AOD range is calculated as the difference between the highest and lowest region-mean values, which in all regions considered turns out to be the difference between MODIS Terra and MISR respectively.

*Author contributions.* RARD led the study design, analysis, and writing, with support and supervision from KvS, AHM, and NPG. JL provided data and developed the components of CanAM-PAM that allow the refractive index of BC to be varied.

*Competing interests.* The authors declare that they have no conflict of interest.

*Acknowledgements.* The authors thank Jérôme Yon for providing spectrally-resolved data for the Bescond et al. (2016) refractive index, and Jason Cole and two anonymous reviewers for their comments on the manuscript. This work has been supported by the Natural Sciences and Engineering Research Council of Canada (NSERC): grants RGPIN-2019-204986 to A.M. and RGPIN-2017-04043 to N.G., and a CGS-D award to R.D.

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
