# Peer review of "The impact of uncertainty in black carbon's refractive index on simulated optical depth and radiative forcing"

_EGUsphere, 2024_

## Referee Comment (RC2)

In this study, the authors investigate the effect of changing the (complex) refractive index of black carbon (BC) aerosol on the aerosol absorption optical depth and on the effective radiative forcing from BC-radiation interactions. They isolate the effect of the BC refractive index by running ensembles of simulations in the atmospheric model CanAM5.1-PAM and varying only the BC refractive index among three values commonly used in Earth system models. The authors state that no previous study has isolated the impact of the refractive index of BC in this way, using a single model with other aerosol treatments held constant and the simulations being otherwise identical. If so, then this is an important assessment to have conducted. The manuscript is well written, and the results are important. As such, in my opinion, this manuscript is appropriate for publication in Atmospheric Chemistry and Physics pending some clarifications and small corrections, as I list below.

1.  It is mentioned in Appendix A, but the authors should also mention in the main part of the text whether the radiative properties of the BC aerosol are affected by humidity.
2.  lines 51-31: "For instance, measurements of scattering and absorption by flame-generated particle may be fit to Mie theory (assuming spherical particles) or Rayleigh-Debye-Gans theory (assuming aggregates)." – Rayleigh-Debye-Gans theory is only approximate, and thus the authors should mention other theoretical frameworks that have been used to fit/model scattering and absorption by BC aggregates, e.g., the multiple sphere T-matrix method (MSTM), the discrete dipole approximation (DDA), and the generalized multi-particle Mie method (GMM). (See for example, Liu and Mishchenko, (2005, 2007), Liu et al. (2008), Sorensen et al. (2018), Kahnert and Kanngießer (2020), and Haspel et al. (2023).) Also, "flame-generated particle" should be "flame-generated particles".
3.  lines 99-100: Why is BC set to 1850 levels in the perturbed ensemble? That sounds more like what a control ensemble would be.
4.  line 227, lines 266-267: Can the authors explain why the impact on total BC effective radiative forcing is not statistically significant even though impact on the effective radiative forcing from BC-radiation interactions is large?
5.  lines 282-291: Even higher values for the complex refractive index of BC have been measured and used in previous studies (e.g., Janzen, 1979; soot G of Fuller et al., 1999; Liu and Mishchenko, 2005, 2007; Liu et al., 2008; Moteki et al., 2010). The authors should mention these and also assess the impact of one of these even higher values.

References
Fuller, K. A., W. C. Malm, and S. M. Kreidenweis, Effects of mixing on extinction by carbonaceous particles, J. Geophys. Res., 104(D13), 15941-15954, 1999.
Haspel, C., C. Zhang, M. J. Wolf, D. J. Cziczo, and M. Sela, Measurements and calculations of enhanced side- and back-scattering of visible radiation by black carbon aggregates, Atmos. Chem. Phys., 23(27), 10091-10115, 2023.
Janzen, J., The refractive index of colloidal carbon, Journal of Colloid and Interface Science, 69, 436-447, 1979.
Kahnert, M., and F. Kanngießer, Modelling optical properties of atmospheric black carbon aerosols, J. Quant. Spectrosc. Radiat. Transf., 244, 106849, 2020.
Liu, L., and M. I. Mishchenko, Effects of aggregation on scattering and radiative properties of soot aerosols, J. Geophys. Res., 110(D11211), doi:10.1029/2004JD005649, 2005.
Liu, L., and M. I. Mishchenko, Scattering and radiative properties of complex soot and soot-containing aggregate particles, J. Quant. Spectrosc. Radiat. Transf., 106, 262-273, 2007.

Liu, L., M. I. Mishchenko, and W. P. Arnott, A study of radiative properties of fractal soot aggregates using the superposition T-matrix method, J. Quant. Spectrosc. Radiat. Transf., 109, 2656-2663, 2008.

Moteki, N., Y. Kondo, and S. Nakamura, Method to measure refractive indices of small nonspherical particles: Application to black carbon particles, J. Aerosol Sci., 41, 513-521, 2010.

Sorensen, C. M., J. Yon, F. Liu, J. Maughan, W. R. Heinson, and M. J. Berg, Light scattering and absorption by fractal aggregates including soot, J. Quant. Spectrosc. Radiat. Transf., 217, 459-473, 2018.

---

## Author Comment (AC1)

This paper attempts to evaluate the impact of the uncertainty of black carbon refractive index on optical thickness and radiative forcing effects, which is a very valuable topic. However, the paper fails to thoroughly discuss the impact of the uncertainty of refractive index. Here are my comments:

We thank the reviewer for their very helpful feedback on our manuscript. Our responses to their comments are provided in green, below.

(1) This paper only tests three refractive indices, whereas in reality, the refractive index of black carbon is far more complex. Firstly, the refractive index of black carbon is spectrally dependent, which the authors have not elaborated on in detail. Secondly, the refractive index of black carbon may vary within a wider range. Even when adopting the maximum refractive index suggested by Bond et al. (2006), the mass absorption cross-section simulated by the current model falls below the lower limit of the observed range. Liu et al. (2020) suggested higher refractive index values by comparing the differences between simulations and measurements.

References:

Bond, T. C., & Bergstrom, R. W. (2006). Light Absorption by Carbonaceous Particles: An Investigative Review. Aerosol Science and Technology, 40 (1),27-67.

Liu, L., & Mishchenko, M. I. (2005). Effects of aggregation on scattering and radiative properties of soot aerosols. Journal of Geophysical Research: Atmospheres,495 110 (D11).

Liu, F., Yon, J., Fuentes, A., Lobo, P., Smallwood, G. J., & Corbin, J. C. (2020). Review of recent literature on the light absorption properties of black carbon: Refractive index, mass absorption cross section, and absorption function. Aerosol Science and Technology, 54 (1), 33-51.

Kahnert, M. (2010). On the discrepancy between modeled and measured mass absorption cross sections of light absorbing carbon aerosols. Aerosol Science and Technology, 44 (6), 453-460.

Luo, J., Zhang, Y., Wang, F., & Zhang, Q. (2018). Effects of brown coatings on the absorption enhancement of black carbon: a numerical investigation. Atmospheric Chemistry and Physics, 18 (23), 16897–16914.

Regarding the choice of BCRI assessed in this work: we agree with the reviewer that even the "high" Bond and Bergstrom (2006) scheme underestimates BC absorption. Our original manuscript restricted its main analysis to BCRI commonly used in Earth system models, and only touched on the impacts of using more strongly-absorbing schemes in the discussion section. In the revised manuscript, we have moved the Bescond et al. (2016) refractive index, which was recommended by Liu et al. (2020), from the discussion to our main analysis. In the process we have renamed our schemes from "low", "medium", and "high" to "dA1991", "BB2006low", "BB2006high", and "Besc2016" for improved clarity. The Besc2016 results are included in all of our figures. In brief, increasing BC absorption from the dA1991 to BB2006low (BB2006high, Besc2016) BCRI scheme increases global-mean 2015-2019 AAOD by 27% (42%, 59%) and ERFari by 32% (47%, 100%).

We then address a number of other BCRI schemes in the discussion [lines 402-419]:

> We have assessed four BCRI schemes here, but many others exist. As well as the Bescond et al. (2016) scheme assessed here, the review by Liu et al. (2020) highlighted the Williams et al. (2007) values of $m_{635nm}$ = 1.75-1.03i and $E(m_{635nm})$ = 0.365 as being consistent with current estimates of the absorption function. Williams et al. (2007) reported measurements at 635nm and 1310nm and did not extrapolate to other wavelengths, but assuming a relatively flat $E(m)$ through the visible range, this would indicate a degree of absorption somewhere between our BB2006high and Besc2016 schemes. Other estimates which have been widely used in the combustion science literature, such as the Janzen (1979) value of m=2.0-1.0i for all visible wavelengths or the more recent Moteki et al. (2010) $m_{1064nm}$ = (2.26±0.13) – (1.26±0.13)i, also yield $E(m)$ between BB2006high and Besc2016, but with values low enough that they are not recommended by Liu et al. (2020).

*Refractive indices determined from laboratory measurements may not be representative of atmospheric black carbon. For instance, BC generated by a simple, clean laboratory flame will likely have a different temperature history – and thus, different optical and structural properties – than that generated by the more complex sources responsible for most atmospheric BC (Bond and Bergstrom, 2006). Combustion experiments also measure freshly emitted particles, which may have substantial morphological differences from hours-to-days old atmospheric BC. The recent work by Moteki et al. (2023), who measured the refractive indices of atmospheric BC particles sampled during a scientific cruise in the northwest Pacific may be more suitable for use in climate models. By combining their optical measurements with the constraints imposed by the accepted mass absorption cross section of BC, they obtained a range of plausible refractive indices suitable for describing atmospheric BC. Their recommended value, $m_{633nm}$ = 1.95 − 0.96i with $E(m_{633nm})$ = 0.297, also falls between the BB2006high and Besc2016 schemes.*

Regarding the spectral dependence of the refractive index: we have expanded our description of the spectral characteristics of the four BCRI schemes [lines 141-144]:

*At all wavelengths, the dA1991 scheme has the lowest absorption and the Besc2016 the highest. E(m) in the first three schemes increase to both the ultraviolet and infrared, while the Besc2016 scheme decreases slightly to the infrared, but all four are fairly constant through the visible and near-infrared (d'Almeida et al., 1991; Chang and Charalampopoulos, 1990; Bescond et al., 2016).*

We have also clarified in the abstract that we vary the full BCRI spectrum, not only its value at 550nm [line 5]:

*With other parameterizations held constant, changing BC's **spectrally-varying** refractive index from the least- to most-absorbing estimate commonly used in Earth system models...*

Our analysis is focused on 550nm, because that is the wavelength for which Earth system models typically publish aerosol optical data. However, we have included references to a now-published companion manuscript, Li et al. (2024), which examines three of our four BCRI (dA1991, BB2006high, and Besc2016) in an offline radiative transfer model. Their work includes assessment of the impacts of BCRI on the spectral dependence of BC optical properties. We clarify this in lines 146-151:

*Although our experiments vary the BCRI at all wavelengths, our analysis predominantly focuses on the characteristics of these schemes at 550nm. This is the wavelength for which Earth system models typically publish aerosol optical data and for which many satellite retrievals are available. In a complementary analysis, Li et al. (2024) assess the optical properties of BC in the dA1991, BB2006high, and Besc2016 schemes, including dependence on wavelength, particle size, and mixing state. The Li et al. (2024) analysis relies on theoretical calculations and a one-dimensional radiative transfer model, while the work presented here explores the impacts of the BCRI on Earth system model simulations.*

---

## Author Comment (AC2)

**review of egusphere-2024-1796**

In this study, the authors investigate the effect of changing the (complex) refractive index of black carbon (BC) aerosol on the aerosol absorption optical depth and on the effective radiative forcing from BC-radiation interactions. They isolate the effect of the BC refractive index by running ensembles of simulations in the atmospheric model CanAM5.1-PAM and varying only the BC refractive index among three values commonly used in Earth system models. The authors state that no previous study has isolated the impact of the refractive index of BC in this way, using a single model with other aerosol treatments held constant and the simulations being otherwise identical. If so, then this is an important assessment to have conducted. The manuscript is well written, and the results are important. As such, in my opinion, this manuscript is appropriate for publication in Atmospheric Chemistry and Physics pending some clarifications and small corrections, as I list below.

We thank the reviewer for their very helpful comments on our manuscript. Our responses to their feedback are provided in green, below. We also wish to highlight the addition of the Bescond et al. (2016) refractive index to our main analysis.

1.  It is mentioned in Appendix A, but the authors should also mention in the main part of the text whether the radiative properties of the BC aerosol are affected by humidity.

    In recognition of its importance for interpreting the manuscript, we have moved the model description from the appendix to the main text (lines 150-192). We have also expanded the description of the calculation of aerosol optical properties in the model (lines 181-191):

    > *Aerosol optical properties in PAM are determined from pre-computed lookup tables. The tables are generated using Mie theory to determine optical properties as a function of relative humidity, wavelength, and particle size. Although the assumption of spherical particles may be inappropriate for freshly-emitted BC, the majority of the BC simulated by an Earth system model is hours to days old and will be relatively compact and/or internally mixed. Mie theory is thus a reasonable approximation. For internally mixed aerosol an effective refractive index is computed using the Maxwell-Garnett approximation (Wu et al., 2018), which has been demonstrated to describe the optical properties of coated BC better than volume-weighted or core-shell approximations (Adachi et al., 2010; Stevens and Dastoor, 2019).*

    > *Other treatments of aerosol morphology and mixing state would likely yield different absolute values of simulated AAOD and ERFari. However, the focus of this work is on the difference in these quantities between simulations conducted with different BCRI schemes, with other parameterizations held constant. As it is, CanAM5.1-PAM's simulated AAOD is in good agreement with other Earth system models and with observations…*

    References:

    Adachi, K., Chung, S. H., and Buseck, P. R.: Shapes of soot aerosol particles and implications for their effects on climate, Journal of Geophysical Research: Atmospheres, 115, https://doi.org/10.1029/2009JD012868, 2010.

    Stevens, R. and Dastoor, A.: A review of the representation of aerosol mixing state in atmospheric models, Atmosphere, 10, https://doi.org/10.3390/atmos10040168, 2019.

    Wu, K., Li, J., von Salzen, K., and Zhang, F.: Explicit solutions to the mixing rules with three-component inclusions, Journal of Quantitative Spectroscopy and Radiative Transfer, 207, 78–82, https://doi.org/10.1016/j.jqsrt.2017.12.020, 2018.

2.  lines 51-31: "For instance, measurements of scattering and absorption by flame-generated particle may be fit to Mie theory (assuming spherical particles) or Rayleigh- Debye-Gans theory (assuming aggregates)." — Rayleigh-Debye-Gans theory is only approximate, and thus the authors should mention other theoretical frameworks that have been used to fit/model scattering and absorption by BC aggregates, e.g., the multiple sphere T-matrix method (MSTM), the discrete dipole approximation (DDA), and the generalized multi-particle Mie method (GMM). (See for example, Liu and Mishchenko, (2005, 2007), Liu et al. (2008), Sorensen et al. (2018), Kahnert and Kanngießer (2020), and Haspel et al. (2023).) Also, "flame-generated particle" should be "flame-generated particles".

We thank the reviewer for this suggestion, and for catching our typo. We have expanded our description of available optical models (lines 59-74):

> *The choice of optical model can introduce substantial uncertainty into the derived refractive index. Historically, many estimates of the refractive index of black carbon have used Mie theory (Mie, 1908) which provides an exact solution to Maxwell's equations for scattering from spherical particles. However, freshly emitted BC particles are not spherical, but instead consist of fractal-like aggregates of individual monomers. Assuming Mie theory can result in substantial underprediction of these particles' absorption and scattering; furthermore, the inferred BCRI describes a combination of pure BC and the air contained within the aggregates' voids, whereas the objective is to measure the refractive index of pure BC (Bond and Bergstrom, 2006). Improving on Mie theory, a number of optical models for aggregate particles exist. Rayleigh-Debye-Gans (RDG) theory remains the most frequently used due to its simplicity (Liu et al., 2020), but it provides an approximate solution only and does not account for multiple scattering between the monomers, which may lead to underestimation of the mass absorption cross section (Mackowski, 1995; Bond and Bergstrom, 2006). Numerically exact solutions for aggregate particles include the multi-sphere T-matrix (MSTM; Mackowski and Mishchenko, 1996) and generalized multi-particle Mie (GMM; Xu, 1995) methods for aggregates composed of non-overlapping spheres, or the discrete dipole approximation (DDA; Purcell and Pennypacker, 1973; Yurkin and Hoekstra, 2007) for more complex morphologies. For more on these methods, the interested reader is referred to Kahnert and Kanngießer (2020). In practice, however, the RDG approximation is often sufficient for BC since its scattering is so low and other uncertainties are so high (Kahnert and Kanngießer, 2020; Mackowski, 1995). None of the BCRI schemes assessed in this work were derived using these more complex optical models.*

We have also included reference to the inversion techniques used to determine the full spectrum of the BCRI from a set of measurements at discrete wavelengths (lines 54-58):

> *The measurements are then inverted to yield the full spectrally-varying refractive index, for example using the Kramers-Kronig relations (Chang and Charalampopoulos, 1990) or the Drude-Lorentz dispersion relation (Dalzell and Sarofim, 1969; Lee and Tien, 1981). The former method is exact, but requires measurements over a greater range of wavelengths; the latter requires fewer measurements but yields poor results at visible wavelengths (Chang and Charalampopoulos, 1990; Bond and Bergstrom, 2006; Menna and D'Alessio, 1982).*

References:

Bond, T. C. and Bergstrom, R. W.: Light Absorption by Carbonaceous Particles: An Investigative Review, Aerosol Science and Technology, 40, https://doi.org/10.1080/02786820500421521, 2006.

Chang, H.-C. and Charalampopoulos, T. T.: Determination of the wavelength dependence of refractive indices of flame soot, Proceedings of the Royal Society of London. Series A: Mathematical and Physical Sciences, 430, 577–591, https://doi.org/10.1098/rspa.1990.0107, 1990.

Kahnert, M. and Kanngießer, F.: Modelling optical properties of atmospheric black carbon aerosols, Journal of Quantitative Spectroscopy and Radiative Transfer, 244, 106 849, https://doi.org/10.1016/j.jqsrt.2020.106849, 2020

Liu, F., Yon, J., Fuentes, A., Lobo, P., Smallwood, G. J., and Corbin, J. C.: Review of Recent Literature on the Light Absorption Properties of Black Carbon: Refractive Index, Mass Absorption Cross Section, and Absorption Function, Aerosol Science and Technology, 54, https://doi.org/10.1080/02786826.2019.1676878, 2020.

Mackowski, D. W.: Electrostatics analysis of radiative absorption by sphere clusters in the Rayleigh limit: application to soot particles, Applied Optics, 34, 3535–3545, https://doi.org/10.1364/AO.34.003535, 1995.

Mackowski, D. W. and Mishchenko, M. I.: Calculation of the T matrix and the scattering matrix for ensembles of spheres, J. Opt. Soc. Am.595 A, 13, 2266–2278, https://doi.org/10.1364/JOSAA.13.002266, 1996

Menna, P. and D'Alessio, A.: Light scattering and extinction coefficients for soot-forming flames in the wavelength range from 200 nm to 600 nm, Symposium (International) on Combustion, 19, 1421–1428, https://doi.org/https://doi.org/10.1016/S0082-0784(82)80319-6, 1982.

Mie, G.: Beiträge zur Optik trüber Medien, speziell kolloidaler Metallösungen, Annalen der Physik, 330, 377–445, https://doi.org/https://doi.org/10.1002/andp.19083300302, 1908.

Purcell, E. M. and Pennypacker, C. R.: Scattering and Absorption of Light by Nonspherical Dielectric Grains, Astrophysical Journal, 186, 705–714, https://doi.org/10.1086/152538, 1973.

Xu, Y.: Electromagnetic scattering by an aggregate of spheres, Applied Optics, 34, 4573–4588, https://doi.org/10.1364/AO.34.004573, 1995.

Yurkin, M. and Hoekstra, A.: The discrete dipole approximation: An overview and recent developments, Journal of Quantitative Spectroscopy and Radiative Transfer, 106, 558–589, https://doi.org/10.1016/j.jqsrt.2007.01.034, iX Conference on Electromagnetic and Light Scattering by Non-Spherical Particles, 2007.

3.   lines 99-100: Why is BC set to 1850 levels in the perturbed ensemble? That sounds more like what a control ensemble would be.

We have slightly rephrased our description of the control and perturbed experiments (lines 201-203)

*In the control ensemble, all emissions of aerosols and greenhouse gases are transient; in the perturbed ensemble, BC emissions are fixed at 1850 levels while other emissions evolve as in the control scenario.*

and added the following sentence to clarify our methodology (lines 204-206):

*This method of calculating ERF is chosen over the alternative approach in which the control scenario uses preindustrial emissions and the perturbed scenario adds transient emissions of the forcer of interest because the latter method does not account for interactions between species.*

4.   line 227, lines 266-267: Can the authors explain why the impact on total BC effective radiative forcing is not statistically significant even though impact on the effective radiative forcing from BC-radiation interactions is large?

We have expanded the paragraph to read (lines 340-347):

*Varying the BCRI, and thus the absorption, of atmospheric BC is found not to have a statistically significant impact on the aerosol-cloud forcing in this experiment. We did not vary the BCRI within cloud droplets, so the only impact on clouds would be via the impact of changes in atmospheric temperature profiles, discussed below. These changes are found to be small relative to the variability of simulated cloud fields. Similarly, the radiative forcing from albedo changes was not found to vary with BCRI scheme because we did not vary the refractive index of BC deposited on snow and ice. The only change in the total BC ERF was thus from the aerosol-radiation component. For the BB2006low and BB2006high schemes, this change was too small to result in a statistically significant increase in total ERF. However, the Besc2016 scheme led to a statistically significant increase in global-mean BC ERF relative to the dA1991 scheme, from -0.02 W/m2 to +0.24 W/m2.*

5.   lines 282-291: Even higher values for the complex refractive index of BC have been measured and used in previous studies (e.g., Janzen, 1979; soot G of Fuller et al., 1999; Liu and Mishchenko, 2005, 2007; Liu et al., 2008; Moteki et al., 2010). The authors should mention these and also assess the impact of one of these even higher values.

We thank the reviewer for highlighting these additional refractive indices.

Our original manuscript restricted its main analysis to BCRI commonly used in Earth system models, and only touched on the impacts of using more strongly-absorbing schemes in the discussion section. In the revised manuscript, we have moved the Bescond et al. (2016) refractive index, which was recommended by Liu et al. (2020), from the discussion to our main analysis. In the process we have renamed our schemes from "low", "medium", and "high" to "dA1991", "BB2006low", "BB2006high", and "Besc2016" for improved clarity. The Besc2016 results are included in all of our figures. In brief, increasing BC absorption from the dA1991 to BB2006low (BB2006high, Besc2016) BCRI scheme increases global-mean 2015-2019 AAOD by 27% (42%, 59%) and ERFari by  32% (47%, 100%).

We then address a number of other BCRI in the discussion (lines 402-419):

*We have assessed four BCRI schemes here, but many others exist. As well as the Bescond et al. (2016) estimate assessed here, the review by Liu et al. (2020) highlighted the Williams et al. (2007) values of m635nm = 1.75-1.03i and E(m635nm) = 0.365 as being consistent with current estimates of the absorption function. Williams et al. (2007) reported measurements at 635nm and 1310nm and did not extrapolate to other wavelengths, but assuming a relatively flat E(m) through the visible range, this would indicate a degree of absorption somewhere between our BB2006high and Besc2016 schemes. Other estimates which have been widely used in the combustion science literature, such as the Janzen (1979) value of m=2.0-1.0i for all visible wavelengths or the more recent Moteki et al. (2010) m_1064nm = (2.26±0.13) – (1.26±0.13)i, also yield E(m) between BB2006high and Besc2016, but with values low enough that they are not recommended by Liu et al. (2020).*

*Refractive indices determined from laboratory measurements may not be representative of atmospheric black carbon. For instance, BC generated by a simple, clean laboratory flame will likely have a different temperature history – and thus, different optical and structural properties – than that generated by the more complex sources responsible for most atmospheric BC (Bond and Bergstrom, 2006). Combustion experiments also measure freshly emitted particles, which may have substantial morphological differences from hours-to-days old atmospheric BC. The recent work by Moteki et al. (2023), who measured the refractive indices of atmospheric BC particles sampled during a scientific cruise in the northwest Pacific may be more suitable for use in climate models. By combining their optical measurements with the constraints imposed by the accepted mass absorption cross section of BC, they obtained a range of plausible refractive indices suitable for describing atmospheric BC. Their recommended value, m633nm = 1.95 – 0.96i with E(m633nm) = 0.297, also falls between the BB2006high and Besc2016 schemes.*

References:

Bescond, A., Yon, J., Ouf, F.-X., Rozé, C., Coppalle, A., Parent, P., Ferry, D., and Laffon, C.: Soot optical properties determined by analyzing extinction spectra in the visible near-UV: Toward an optical speciation according to constituents and structure, Journal of Aerosol Science, 101, 118–132, https://doi.org/10.1016/j.jaerosci.2016.08.001, 2016.

Janzen, J.: The refractive index of colloidal carbon, Journal of Colloid and Interface Science, 69, 436–447, https://doi.org/10.1016/0021- 9797(79)90133-4, 1979.

Liu, F., Yon, J., Fuentes, A., Lobo, P., Smallwood, G. J., and Corbin, J. C.: Review of Recent Literature on the Light Absorption Properties of Black Carbon: Refractive Index, Mass Absorption Cross Section, and Absorption Function, Aerosol Science and Technology, 54, https://doi.org/10.1080/02786826.2019.1676878, 2020.

Moteki, N., Kondo, Y., and Nakamura, S.-i.: Method to measure refractive indices of small nonspherical particles: Application to black carbon particles, Journal of Aerosol Science, 41, 513–521, https://doi.org/https://doi.org/10.1016/j.jaerosci.2010.02.013, 2010.

Moteki, N., Ohata, S., Yoshida, A., and Adachi, K.: Constraining the complex refractive index of black carbon particles using the complex forward-scattering amplitude, Aerosol Science and Technology, 57, 678–699, https://doi.org/10.1080/02786826.2023.2202243, 2023.

Williams, T., Shaddix, C., Jensen, K., and Suo-Anttila, J.: Measurement of the dimensionless extinction coefficient of soot within laminar diffusion flames, International Journal of Heat and Mass Transfer, 50, 1616–1630, https://doi.org/10.1016/j.ijheatmasstransfer.2006.08.024, 2007.

**References**

Fuller, K. A., W. C. Malm, and S. M. Kreidenweis, Effects of mixing on extinction by carbonaceous particles, J. Geophys. Res., 104(D13), 15941-15954, 1999.

Haspel, C., C. Zhang, M. J. Wolf, D. J. Cziczo, and M. Sela, Measurements and calculations of enhanced side- and back-scattering of visible radiation by black carbon aggregates, Atmos. Chem. Phys., 23(27), 10091-10115, 2023.

Janzen, J., The refractive index of colloidal carbon, Journal of Colloid and Interface Science, 69, 436-447, 1979.

Kahnert, M., and F. Kanngießer, Modelling optical properties of atmospheric black carbon aerosols, J. Quant. Spectrosc. Radiat. Transf., 244, 106849, 2020.

Liu, L., and M. I. Mishchenko, Effects of aggregation on scattering and radiative properties of soot aerosols, J. Geophys. Res., 110(D11211), doi:10.1029/2004JD005649, 2005.

Liu, L., and M. I. Mishchenko, Scattering and radiative properties of complex soot and soot-containing aggregate particles, J. Quant. Spectrosc. Radiat. Transf., 106, 262-273, 2007.
Liu, L., M. I. Mishchenko, and W. P. Arnott, A study of radiative properties of fractal soot aggregates using the superposition T-matrix method, J. Quant. Spectrosc. Radiat. Transf., 109, 2656-2663, 2008.

Moteki, N., Y. Kondo, and S. Nakamura, Method to measure refractive indices of small nonspherical particles: Application to black carbon particles, J. Aerosol Sci., 41, 513-521, 2010.

Sorensen, C. M., J. Yon, F. Liu, J. Maughan, W. R. Heinson, and M. J. Berg, Light scattering and absorption by fractal aggregates including soot, J. Quant. Spectrosc. Radiat. Transf., 217, 459-473, 2018.